# Intra-atrial activation pattern is useful to localize the areas of non-pulmonary vein triggers of atrial fibrillation

**Kazuo Sakamoto**[1], **Yasushi Mukai**[1,2]*, **Shunsuke Kawai**[1,2], **Kazuhiro Nagaoka**[3], **Shujiro Inoue**[4], **Susumu Takase**[1], **Daisuke Yakabe**[1,5], **Shota Ikeda**[1], **Hiroshi Mannoji**[6], **Tomomi Nagayama**[1], **Akiko Chishaki**[7], **Hiroyuki Tsutsui**[1]

1 Department of Cardiovascular Medicine, Kyushu University Graduate School of Medical Sciences, Maidashi, Higashi-ku, Fukuoka, Japan, 2 Department of Cardiovascular Medicine, Japanese Red Cross Fukuoka Hospital, Okusu, Minami-ku, Fukuoka, Japan, 3 Department of Cardiology, St. Mary's Hospital, Tsubuku-honmachi, Kurume, Fukuoka, Japan, 4 Department of Cardiology, Aso Iizuka Hospital, Yoshio-machi, Iizuka, Fukuoka, Japan, 5 Department of Cardiology, Kyushu Medical Center, Jigyohama, Chuo-ku, Fukuoka, Japan, 6 Department of Cardiology, Hamanomachi Hospital, Nagahama, Chuo-ku, Fukuoka, Japan, 7 Division of Cardiology, Fukuoka Dental College Hospital, Tamura, Sawara-ku, Fukuoka, Japan

☯ These authors contributed equally to this work.
* y_mukai@junnai.org

**Data Availability Statement:** Data cannot be shared publicly because of the matter about personal information. Data are available from the Kyushu University Institutional Review Board

## Abstract

### Background

Pulmonary vein isolation (PVI) is an established ablation procedure for atrial fibrillation (AF), however, PVI alone is insufficient to suppress AF recurrence. Non-pulmonary vein (non-PV) trigger ablation is one of the promising strategies beyond PVI and has been shown to be effective in refractory/persistent AF cases. To make non-PV trigger ablation more standardized, it is essential to develop a simple method to localize the origin of non-PV triggers.

### Methods

We retrospectively analyzed 37 non-PV triggers in 751 ablation sessions for symptomatic AF from January 2017 to December 2020. Regarding non-PV triggers, intra-atrial activation interval from the earliest in right atrium (RA) to proximal coronary sinus (CS) (RA-CSp) and that from the earliest in RA to distal CS (RA-CSd) obtained by a basically-positioned duodecapolar RA-CS catheter were compared among 3 originating non-PV areas [RA, atrial septum (SEP) and left atrium (LA)].

### Results

RA-CSp of RA non-PV trigger (56.4 ± 23.4 ms) was significantly longer than that of SEP non-PV (14.8 ± 25.6 ms, p = 0.019) and LA non-PV (-24.9 ± 27.9 ms, p = 0.0004). RA-CSd of RA non-PV (75.9 ± 32.1 ms) was significantly longer than that of SEP non-PV (34.2 ± 32.6 ms, p = 0.040) and LA non-PV (-13.3 ± 41.2 ms, p = 0.0008). RA-CSp and RA-CSd of SEP non-PV were significantly longer than those of LA non-PV (p = 0.022 and p = 0.016, respectively). Sensitivity and specificity of an algorithm to differentiate the area of non-PV

(ijkseimei@jimu.kyushu-u.ac.jp) on reasonable request.

**Funding:** The author(s) received no specific funding for this work

**Competing interests:** The authors have declared that no competing interests exist.

trigger using RA-CSp (cut-off value: 50 ms) and RA-CSd (cut-off value: 0 ms) were 88% and 97% for RA non-PV, 81% and 73% for SEP non-PV, 65% and 95% for LA non-PV, respectively.

## Conclusions

The analysis of intra-atrial activation sequences was useful to differentiate non-PV trigger areas. A simple algorithm to localize the area of non-PV trigger would be helpful to identify non-PV trigger sites in AF ablation.

## Introduction

Pulmonary vein isolation (PVI) is an established ablation procedure for atrial fibrillation (AF), however, PVI alone is insufficient to suppress recurrences of AF [1–4]. In order to achieve better outcomes, additional ablation procedures beyond PVI have been explored [5]. Several ablation procedures such as left atrial posterior wall isolation, linear ablation, ganglionated plexus ablation, complex fractionated atrial electrogram (CFAE) ablation, low voltage area (LVA) ablation and non-pulmonary vein (non-PV) trigger ablation have been reported to be effective, however, none of them have become an established procedure beyond PVI so far [6–10]. While linear-based ablations including posterior wall isolation are uniform procedures, CFAE ablation, LVA ablation and non-PV trigger ablation may be unique treatments for each patient. Recently, it has been reported that elimination of non-PV triggers is related to favorable outcomes in persistent AF as well as in paroxysmal AF [11–15]. Although the impact of non-PV triggers in AF has become more evident, non-PV trigger ablation has not been standardized as an additional procedure to PVI, partly because of the need of skilled catheter technique of non-PV trigger mapping [16].

The aim of this study was to compare intra-atrial activation sequences of non-PV triggers from different areas of the atria, and to establish simple criteria for localizing the areas of non-PV triggers using basic electrodes positioned by default.

## Methods

### Patient population

We retrospectively analyzed 37 non-PV triggers in 751 ablation sessions for symptomatic AF (including 310 paroxysmal AF) from January 2017 to December 2020. This study was in compliance with the principles outlined in the Declaration of Helsinki and was approved by the institutional review board for ethics at our institution, Kyushu University Hospital (approval no. 29–44). Informed consent was obtained in the form of opt-out on the web-site (https://www.cardiol.med.kyushu-u.ac.jp/research/clinical-research/).

### Electrophysiological study and ablation of non-PV trigger

Antiarrhythmic drugs were discontinued at least five half-lives before AF ablation. All patients underwent electrophysiological study (EPS) and catheter ablation under deep sedation and fasting conditions. A 20-pole catheter (BeeAT, Japan Lifeline, Tokyo, Japan) was inserted via right jugular vein, and the proximal portion of the catheter was placed along the superior vena cava (SVC) and crista terminalis (CT) and the distal portion was located in the coronary sinus (CS) (Fig 1A). Another 10-pole catheter was inserted for right ventricular pacing. Following

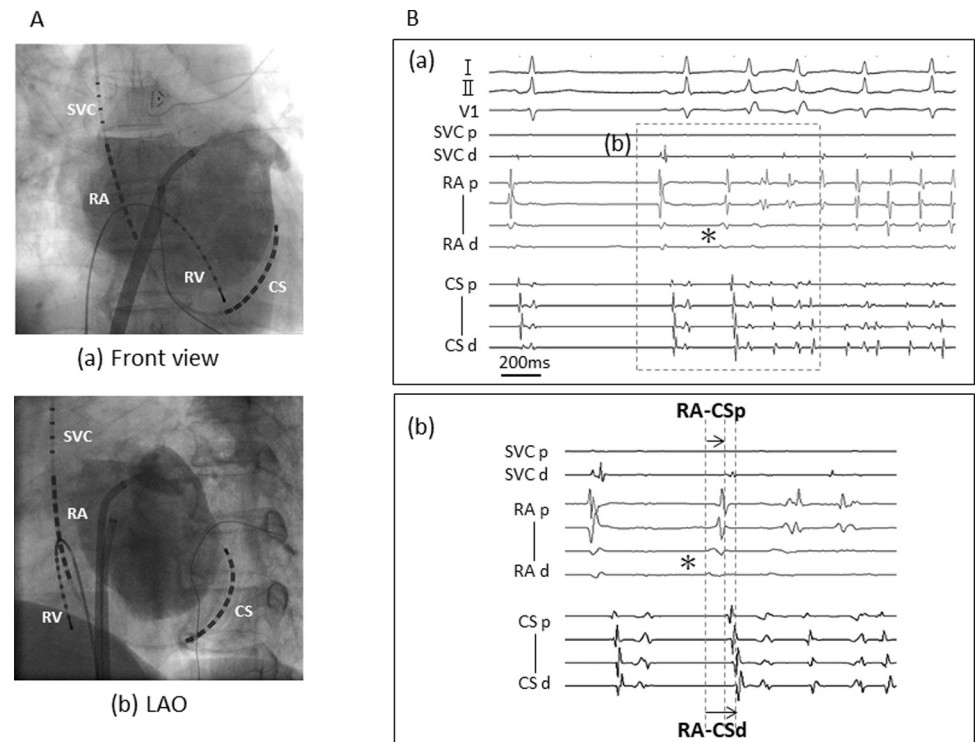

**Fig 1. Catheter position and intra-cardiac electrograms of AF initiation by a non-PV trigger.** A, basically-positioned RA-CS catheter and RV catheter in front view (a) and LAO (b). B, Surface electrocardiograms and intra-cardiac electrograms of AF initiation by a non-PV trigger (a) and focused intra-cardiac electrograms of the premature atrial beat of a non-PV trigger indicating intra-cardiac activation intervals analyzed in the present study (b). RA-CSp and RA-CSd are the intra-atrial activation interval from the earliest in RA to proximal CS and that from the earliest in RA to distal CS, respectively. RA, right atrium; CS, coronary sinus; RV, right ventricle; LAO, left anterior oblique; SVC, superior vena cava.

trans-septal puncture under guidance of intra-cardiac echocardiography (5.5 ~ 10 MHz, 8 Fr, AcuNav, Biosense Webster, Diamond Bar, CA), two long sheaths (SL1, AF Division, Abbott/ St. Jude Medical, Minneapolis, MN, USA) were introduced into left atrium (LA). We used CARTO system (Biosense Webster, Diamond Bar, CA, USA) or Ensite system (Abbott, St. Paul, MN, USA) with catheters corresponding to either 3D mapping system. That is, CARTO system used Lasso, PENTARAY Nav catheter and Navistar THERMOCOOL (Biosense Webster, Diamond Bar, CA, USA), while Ensite system used a 20-pole circular catheter, HD Grid and TactiCath (Abbott, St. Paul, MN, USA).

Regardless of the first or multiple sessions, we completed extended PVI. After completion of PVI, we performed non-PV trigger induction test in all sessions. The induction and identification methods were as follows. When spontaneous AF initiation was not observed, we tried to induce AF in one or both the following ways: (1) induction with a bolus injection of isoproterenol (3–5 μg) and adenosine (20 mg) [15]. If not induced, we induced AF by atrial burst pacing (30-beat at an amplitude of 10V and pulse width of 1 ms from the ostium of CS; increasing from 240 to 320 ppm in steps of 20 ppm) with a bolus injection of isoproterenol (3–5 μg). Then, we conducted intra-cardiac cardioversion (10–30 J) with BeeAT and restored AF to sinus rhythm. After termination of AF, we observed if AF was spontaneously initiated for a few minutes. Once AF was initiated, we focused on the premature atrial beat (PAB) that initiated AF and we defined the PAB as a non-PV trigger. Multipolar mapping catheters (PENTARAY Nav/HD Grid and circular catheters) were repositioned to the originating area of the

non-PV trigger according to the intra-atrial activation sequences. Again, the trigger PAB was mapped and localized regarding the prematurity of electrograms recorded in the multipolar mapping catheters [17, 18]. When the prematurity was confined in the mapping catheters, we ablated the earliest point as the origin of AF trigger. Finally, the localization of the non-PV trigger was confirmed when the PAB initiating AF was eliminated after ablation using the aforementioned induction protocol. AF induction test was repeated to see if there were any other non-PV triggers remaining.

## Analysis of intra-atrial activation pattern of non-PV trigger

Non-PV triggers that appeared three times or more were analyzed and triggers from SVC were excluded. Intra-atrial activation interval from the earliest in right atrium (RA) to proximal CS (RA-CSp) and that from the earliest in RA to distal CS (RA-CSd) in a BeeAT catheter were measured in each non-PV trigger. The earliest in RA was defined as the first deflection from baseline in RA electrodes (Fig 1B). Both RA-CSp and RA-CSd for each trigger were averaged over a series of initiations.

RA-CSp and RA-CSd were compared among three originating areas of non-PV triggers; RA, atrial septum (SEP) and LA. Diagnostic thresholds of RA-CSp and RA-CSd were calculated to distinguish the three areas (RA, SEP and LA) in the following ways. Finally, a simple algorithm to differentiate the areas of the origins was created by combining the calculated thresholds of RA-CSp and RA-CSd.

## Statistical analysis

The data are expressed as mean ± standard deviation (SD) for continuous variables, and numbers and percentages for categorical variables. A comparison of continuous variables and categorical variables between pairs of groups was carried out using the Kruskal–Wallis test with the Steel–Dwass post hoc test and the chi-square test or Fisher's exact test, appropriately. The thresholds of RA-CSp and RA-CSd were determined based on the area under the curve (AUC) of the receiver operating characteristic (ROC) curve. Moreover, the sensitivity and specificity of the algorithm differentiating non-PV triggers were verified. JMP software (Ver. 15, SAS institute Inc., NC, USA) was used for all statistical analyses. The significance of the comparison between groups was defined as $p < 0.05$.

## Results

### Patient characteristics and catheter ablation

Patient characteristics of 35 patients with 37 non-PV triggers are shown in Table 1. Mean age was 65.7 ± 10.1 years, and 25 (71%) were male. Twelve sessions were first sessions while 23 sessions were of multiple sessions. There were 17 sessions (49%) with paroxysmal and 18 sessions (51%) with persistent AF. Five out of 35 sessions had undergone cavotricuspid isthmus (CTI) block line in previous sessions. Left ventricular ejection fraction was 64.9 ± 8.1% and left atrial dimension was 41.9 ± 6.8 mm. The mean CHADS2 score was 1.1±1.1. Number of studied non-PV triggers in RA, SEP and LA were 8, 11 and 16, respectively. There were no significant differences in patient characteristics among the three areas. We considered supraventricular tachycardias (SVT) as possible triggers of AF as well [19]. CTI-dependent atrial flutters were induced during EPS in 3 cases, but no other SVTs were induced in the studied patients.

The 37 non-PV triggers were successfully mapped and eliminated after 5.5 ± 2.5 times of induction and 4.4±3.5 times of cardioversion. There were no significant differences among three areas regarding these numbers of times (Table 2).

**Table 1. Patient characteristics.**

| Variable | All (n = 35) | RA (n = 8) | SEP (n = 11) | LA (n = 16) | P value |
|---|---|---|---|---|---|
| Age (years) mean±SD | 65.7±10.1 | 67.1±4.1 | 63.1±9.5 | 66.7±12.4 | 0.486 |
| Male, n (%) | 25 (71) | 5(63) | 9 (81) | 11(69) | 0.610 |
| PAF, n (%) | 17 (49) | 5 (63) | 4 (36) | 8 (50) | 0.521 |
| Multi-sessions, n (%) | 23 (66) | 4 (50) | 3 (27) | 5 (31) | 0.563 |
| Previous CTI block line, n (%) | 5 (14) | 0 (0) | 2 (18) | 3 (19) | 0.421 |
| LVEF (%) mean±SD | 64.9±8.1 | 65.6±4.7 | 63.8±10.6 | 65.4±7.9 | 0.940 |
| LAD (%) mean±SD | 41.9±6.8 | 43.4±7.6 | 39.2±8.4 | 43.1±4.7 | 0.561 |
| Hypertension, n (%) | 22 (63) | 5 (63) | 9 (82) | 8 (50) | 0.225 |
| Diabetes mellitus, n (%) | 4 (11) | 1 (13) | 1 (9) | 2 (13) | 0.956 |
| Heart failure, n (%) | 5 (14) | 1 (13) | 3 (27) | 1 (6) | 0.315 |
| Stroke, n (%) | 2 (6) | 1 (13) | 0 (0) | 1 (6) | 0.402 |
| CHADS2 (points) mean±SD | 1.1±1.1 | 1.1±1.2 | 1.2±0.8 | 1.1±1.3 | 0.856 |

RA, right atrium; SEP, atrial septum; LA, left atrium; PAF, paroxysmal atrial fibrillation; CTI, cavotricupid isthmus; LVEF, left ventricular ejection fraction; LAD, left atrial diameter

## Intra-atrial activation interval of non-PV triggers

RA-CSp of RA non-PV trigger was 56.4 ± 23.4 ms, which was longer than that of SEP non-PV trigger (14.8 ± 25.6 ms, p = 0.019) and LA non-PV trigger (-24.9 ± 27.9 ms, p = 0.0004) (Fig 2A). RA-CSd of RA non-PV trigger was 75.9 ± 32.1 ms and was significantly longer than SEP non-PV trigger (34.2 ± 32.6 ms, p = 0.040) and LA non-PV trigger (-13.3 ± 41.2 ms, p = 0.0008) (Fig 2B). RA-CSp of SEP non-PV trigger was significantly longer than that of LA non-PV trigger (p = 0.022) as well as in RA-CSd trigger (p = 0.016).

## Creating an algorithm to localize the area of non-PV trigger

Two ROC curves were created (Fig 3): one is regarding RA-CSp to distinguish RA non-PV trigger from SEP and LA non-PV trigger (A), and the other is RA-CSd to distinguish SEP non-PV trigger from LA non-PV trigger after exclusion of RA non-PV trigger (B). The threshold of RA-CSp was 50 ms with AUC of 0.938 and RA-CSd was 1 ms with AUC of 0.809. Their sensitivity and specificity of RA-CSp were 0.88 and 0.96, and those of RA-CSd were 0.91 and 0.65, respectively.

An algorithm to differentiate the area of non-PV trigger was developed by incorporating the two cut-off values of intra-atrial interval; RA-CSp of 50 ms and RA-CSd of 0 ms (Fig 4). Diagnostic accuracy of this algorithm in the present 37 non-PV triggers was validated. Sensitivity and specificity for RA non-PV trigger were 88% and 97%, respectively; for SEP non-PV, 81% and 73%, respectively; for LA non-PV, 65% and 95%, respectively (Table 3).

**Table 2. Incidences of non-PV trigger firing and the number of cardioversion for AF.**

| Variable | All (n = 37) | RA (n = 8) | SEP (n = 12) | LA (n = 17) | P value |
|---|---|---|---|---|---|
| The incidence of AF firing | 5.5±2.5 | 4.8±3.1 | 5.6±3.5 | 5.8±1.9 | 0.448 |
| The number of cardioversion for AF | 4.4±3.5 | 3.4±2.1 | 4.9±4.9 | 4.5±2.9 | 0.657 |

RA, right atrium; SEP, atrial septum; LA, left atrium

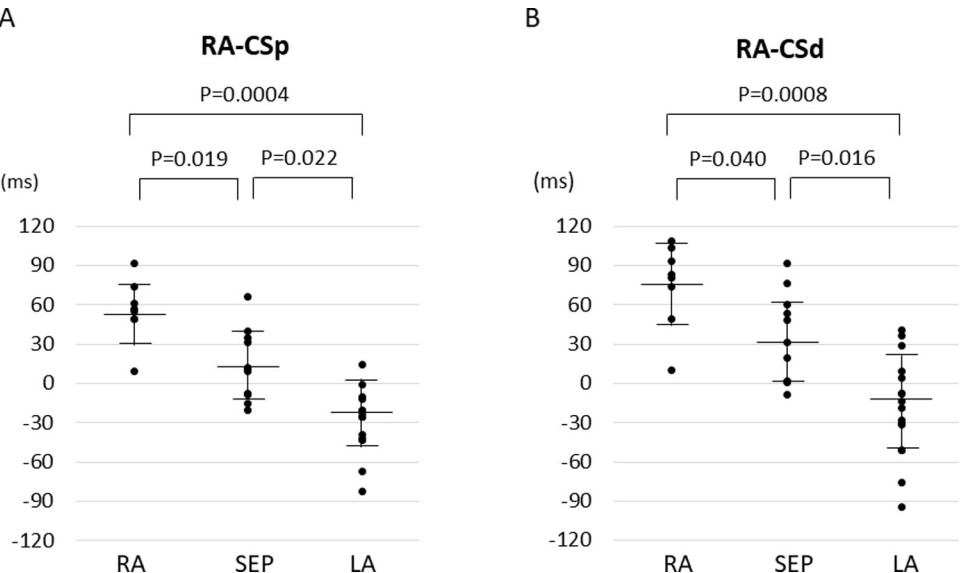

**Fig 2. Comparison of intra-atrial activation intervals of non-PV trigger among RA, SEP and LA.** RA-CSp, intra-atrial activation interval from the earliest in RA to proximal CS; RA-CSd, intra-atrial activation interval from the earliest in RA to distal CS; RA, right atrium; SEP, atrial septum; LA, left atrium; CS, coronary sinus.

## Discussion

In the present study, it was demonstrated that analyzing intra-atrial activation sequences obtained by a basically-positioned duodecapolar RA-CS catheter was useful to differentiate non-PV trigger area among RA, SEP and LA. We could create a simple algorithm to localize the area of non-PV trigger using intra-atrial activation intervals with an acceptable diagnostic accuracy.

### Clinical importance of non-PV trigger mapping and ablation

Several ablation procedures such as linear-based ablations, LVA ablation and non-PV trigger ablation have been explored to achieve the better outcome beyond PVI, however, none of them has become an established procedure beyond PVI. The efficacy of linear-based ablation has been inconsistent among studies, which may be due to the different degree of atrial degeneration and the various mechanisms of AF in every patient [4, 20]. In contrast, non-PV trigger ablation is a unique approach for each patient and could theoretically target every AF initiation [12, 21]. We previously reported that ablation outcome in AF patients with non-PV triggers could be improved to the level of those without non-PV triggers, by selective mapping and ablation of non-PV triggers [15]. For detailed mapping, it is also important to evaluate the substrate such as fibrosis by using multipolar electrodes [22]. Therefore, a standardized mapping strategy for non-PV trigger mapping has been awaited.

The prevalence of non-PV trigger in AF ablation has been reported from 5% to 30%, however, non-PV trigger was detected in 35 among 751 sessions (4.7%) in this study and its rate was lower than the previous studies [12, 23, 24]. This may be related to the different induction manners such as the dose of drugs and the conditions of the patients [25–27]. Furthermore, the nonuniform definition of non-PV trigger among studies could be another reason [12, 23, 24]. The present study excluded the trigger in SVC because a PAB from SVC was easily discriminated with the intra-atrial activation pattern [28]. Moreover, PABs that induced AF less than three times and bare PABs that did not initiate AF were also excluded. In terms of the

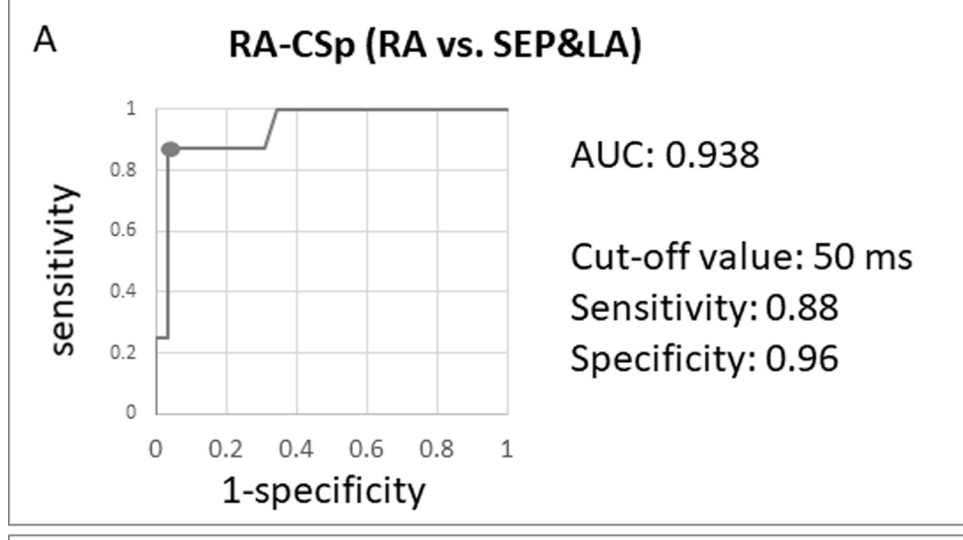

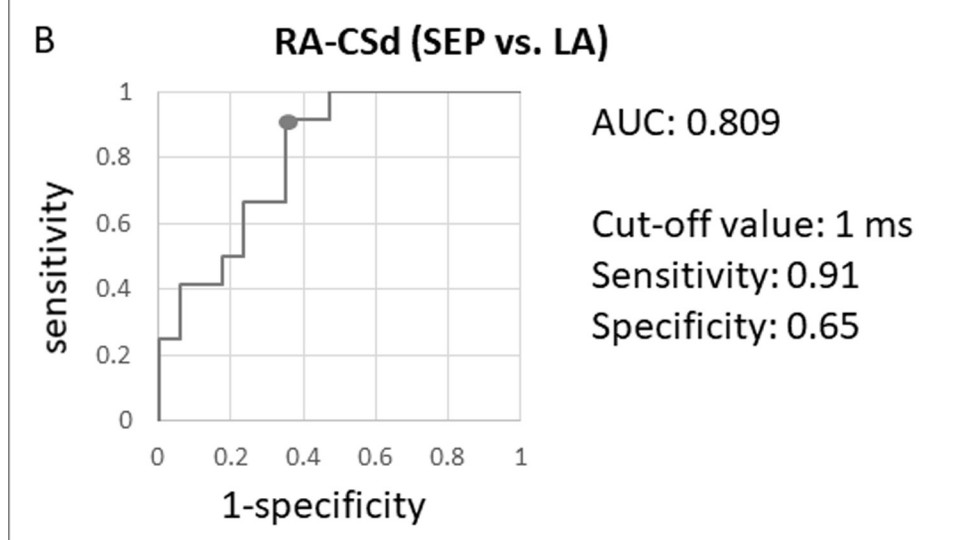

**Fig 3. Receiver operating characteristic curves and cut-off value to distinguish the areas of non-PV trigger.** A. To distinguish RA from SEP and LA with RA-CSp. B, To distinguish SEP from LA with RA-CSd. RA-CSp, intra-atrial activation interval from the earliest in RA to proximal CS; RA-CSd, intra-atrial activation interval from the earliest in RA to distal CS; RA, right atrium; CS, coronary sinus; SEP, atrial septum; LA, left atrium; AUC, area under the curve.

originating area of non-PV triggers, RA, SEP and LA accounted for 22%, 32% and 46%, respectively, and its distribution was similar to previous reports [24, 28]. The detailed localization was CT in 14%, right atrial septum in 16%, left atrial septum in 16%, left atrial posterior wall in 35% and left atrial anterior wall in 5.4%.

The occurrence of non-PV triggers was comparable in paroxysmal (17 in 310, 5.5%) and in non-paroxysmal cases (18 in 441, 4.1%). The Prevalence of non-PV triggers was similar regardless of arrhythmia type, which is consistent to previous reports including ours [15, 24].

## The stepwise mapping strategy to identify the non-PV trigger

The mapping procedure of non-PV trigger is a time-consuming and complicated procedure. The stepwise strategy to localize the non-PV trigger includes a general estimation of the non-

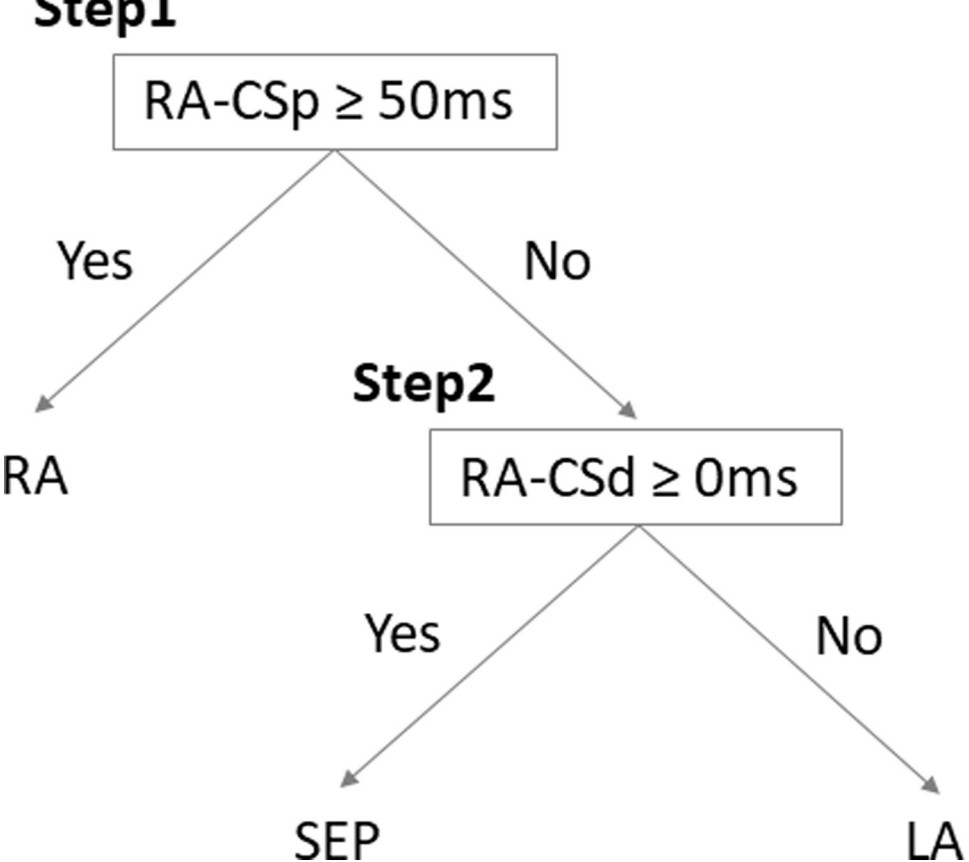

**Fig 4. An algorithm to differentiate the areas of non-PV triggers.** RA-CSp, intra-atrial activation interval from the earliest in RA to proximal CS; RA-CSd, intra-atrial activation interval from the earliest in RA to distal CS; RA, right atrium; SEP, atrial septum; LA, left atrium; CS, coronary sinus.

PV trigger area and a detailed search thereafter. As the first step, we estimated the area of non-PV trigger by intra-atrial activation sequences obtained by a basically-positioned duodecapolar RA-CS catheter. For the second step, we searched for the detailed localization of the trigger using multipolar mapping catheters (PENTARAY Nav / HD Grid) [15, 18]. Our stepwise strategy uncovered and eliminated all the non-PV triggers whereas the initial estimation of non-PV trigger area is important at the beginning. Kubala et al. reported the efficacy of the localization algorithm for non-PV trigger with a combination of surface electrocardiogram and intra-cardiac electrograms, however, interpretation of ectopic P wave morphology was sometimes challenging during AF ablation [29]. On the other hand, intra-atrial activation sequences obtained by a basically-positioned duodecapolar RA-CS catheter in the present study could be

**Table 3. Accuracy of the algorithm to differentiate the area of non-PV triggers.**

|     | Sensitivity (%) | Specificity (%) | PPV (%) | NPV (%) |
| --- | --- | --- | --- | --- |
| RA  | 88 | 97 | 88 | 97 |
| SEP | 91 | 73 | 59 | 95 |
| LA  | 65 | 95 | 92 | 76 |

RA, right atrium; SEP, atrial septum; LA, left atrium; PPV, positive predictive value; NPV, negative predictive value

easily measured and utilized for estimating non-PV trigger sites. Thus, we aimed to quantitate the activation sequences and validated the predicting accuracy by them.

### Intra-atrial activation sequences of non-PV triggers in three different areas

We compared intra-atrial activation sequences of non-PV triggers in three areas including RA, SEP and LA to solve the practical problem of where to place multipolar mapping catheters for detailed localization of the triggers. We found that two intra-atrial activation intervals, RA-CSp and RA-CSd, were different among three areas (Fig 2). A non-PV trigger originating from RA activated RA in prior to CS, that was, RA-CSp and RA-CSd was fully positive. In SEP, RA-CSp could be positive or negative depending on whether the origin was left or right side of atrium. As for LA non-PV trigger, distal CS preceded proximal CS and RA, resulting in a greatly negative in RA-CSd compared to RA-CSp. Thus, differences of the two indices among three areas were reasonable from the perspective of anatomical theory. In addition, recording the local electrograms using a basically-positioned single catheter rather than multiple catheters generalized the inter-electrode distances and made the analysis of intra-atrial activation intervals possible.

We advocated a simple algorithm that could differentiate the areas of non-PV triggers utilizing two indices, RA-CSp and RA-CSd, with sufficient accuracy. This algorithm alone cannot localize the detailed origin of the trigger, however, it may be useful as a preliminary step before a detailed mapping with multipolar catheters. In other words, it could shorten the time of non-PV trigger mapping and could be useful to standardize the technique for non-PV trigger ablation [30].

### Limitations

First, this study was a retrospective study analyzing a limited sample size, therefore, a prospective study with more data is needed to standardize this analysis method for clinical practice. Also, the algorithm estimating the area of non-PV trigger needs to be verified using external validation data.

Second, it was uncertain whether all the non-PV triggers were uncovered and eliminated in the real practice. The low incidence in this study of non-PV triggers may be partly because we excluded SVC triggers in this study. Another reason may be because non-PV triggers were induced after completion of PVI in this study. Thus, non-PV triggers originating from near PV antrum might have been eliminated by PVI. Especially, we excluded frequent PABs without AF and PABs that initiated AF less than three times and the amount of non-PV triggers eliminated in this study might be smaller as compared to previous ones.

Third, despite this study with limited cases could not mention the impact of conduction block lines such as CTI block line, lateral mitral isthmus line or LVA block, patients with previous ablation or LVA should be carefully analyzed.

Fourth, the assessment of the intra-cardiac electrograms might be difficult soon after a defibrillation because it takes a few seconds for the baseline of electrograms to return to where it was.

Finally, the electrode catheter used in this study, which can simultaneously record CS and RA potentials (BeeAT), is not available worldwide and can only be used in limited regions. Therefore, it is necessary to verify whether the same results can be obtained using widely available separate CS and RA catheters.

### Conclusions

Intra-atrial activation sequences of non-PV triggers obtained by a basically-positioned duodecapolar RA-CS catheter were different among RA, SEP and LA and were useful to estimate the

area of non-PV trigger. We proposed a simple algorithm to localize the areas of non-PV triggers, which was accurate and might be helpful to identify non-PV triggers in AF ablation.

## Author Contributions

**Data curation:** Shunsuke Kawai, Kazuhiro Nagaoka.

**Writing – original draft:** Kazuo Sakamoto.

**Writing – review & editing:** Yasushi Mukai, Shujiro Inoue, Susumu Takase, Daisuke Yakabe, Shota Ikeda, Hiroshi Mannoji, Tomomi Nagayama, Akiko Chishaki, Hiroyuki Tsutsui.

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
