## [Decision Letter · Decision Letter 0]

6 Dec 2021

PONE-D-21-34225Intra-atrial activation pattern is useful to localize the areas of non-pulmonary vein triggers of atrial fibrillationPLOS ONE

Dear Dr. Mukai,

Thank you for submitting your manuscript to PLOS ONE. After careful consideration, we feel that it has merit but does not fully meet PLOS ONE’s publication criteria as it currently stands. Therefore, we invite you to submit a revised version of the manuscript that addresses the points raised during the review process.

ACADEMIC EDITOR: Dear Authors, after a careful evaluation of your paper, according to our reviewers' suggestion, our opinion is that your manuscript cannot be accepted as it is, but it needs major revisions. In particular, some methodological issues needs to be better clarified. 

We look forward to receiving your revised manuscript.

Kind regards,

Luigi Sciarra

Academic Editor

PLOS ONE

Journal Requirements:

Additional Editor Comments:

Dear Authors, after a careful evaluation of your paper, according to our reviewers' suggestion, our opinion is that your manuscript cannot be accepted as it is, but it needs major revisions.

Reviewers' comments:

Reviewer's Responses to Questions

**Comments to the Author**

1. Is the manuscript technically sound, and do the data support the conclusions?

Reviewer #1: Yes

Reviewer #2: Yes

Reviewer #3: Yes

2. Has the statistical analysis been performed appropriately and rigorously? 

Reviewer #1: Yes

Reviewer #2: N/A

Reviewer #3: Yes

3. Have the authors made all data underlying the findings in their manuscript fully available?

Reviewer #1: Yes

Reviewer #2: Yes

Reviewer #3: Yes

4. Is the manuscript presented in an intelligible fashion and written in standard English?

Reviewer #1: Yes

Reviewer #2: Yes

Reviewer #3: Yes

5. Review Comments to the Author

Reviewer #1: The work is very interesting and well written.

I would discuss two aspects in depth:

1. Before the analysis of non-PV triggers, the importance of the electrophysiological study, especially in paroxysms without heart disease, should be highlighted in order to find any organized arrhythmias. Please read Sciarra L, Rebecchi M, De Ruvo E, De Luca L, Zuccaro LM, Fagagnini A, Corò L, Allocca G, Lioy E, Delise P, Calò L. How many atrial fibrillation ablation candidates have an underlying supraventricular tachycardia previously unknown? Efficacy of isolated triggering arrhythmia ablation. Europace. 2010 Dec;12(12):1707-12. doi: 10.1093/europace/euq327. Epub 2010 Sep 10. PMID: 20833693.

2. I would also briefly mention the role of the substrate as the islets of fibrosis in the left atrium (Rillo M, Palamà Z, Punzi R, Vitanza S, Aloisio A, Polini S, Tucci A, Pollastrelli A, Zonno F, Anastasia A, Giannattasio CF, My L. A new interpretation of nonpulmonary vein substrates of the left atrium in patients with atrial fibrillation. J Arrhythm. 2021 Feb 22;37(2):338-347. doi: 10.1002/joa3.12521. PMID: 33850575; PMCID: PMC8021999. )

Reviewer #2: Comments to the Author

This paper focused on the method to distinguish the areas of non-pulmonary vein triggers of AF with an electrode catheter.

I clinically agree with author’s ideas, but there are some major problems.

In particular, parts of the Methods section are considerably lacking.

Major comments

1. This CS catheter you recommend is generally difficult to use. Aren’t the electrode catheters over-used, such as the PentaRay or BeeAT catheter? Therefore, I think it is difficult to use this method as a standardized algorithm.

2. Although there is a description of 751 sessions, the total number of cases and paroxysmal and non-paroxysmal cases are unknown. This study is a retrospective study and more patient information should be provided.

3. Were the 35 cases in this study completely cured for the entire follow-up period by the focal ablation of the non-pulmonary vein origins? Did you try to re-induce AF after the ablation of the non-pulmonary vein origins? This study found that paroxysmal and non-paroxysmal non-pulmonary vein origins were approximately equal. Moreover, in the intracardiac electrocardiogram shown in Fig. 1B, the onset of the AF is far from the previous QRS. From the above, it is hard to believe that it was the correct site to prove the origin of AF. Finally, I consider it to have an overall low credibility.

4. There were too few cases of non-pulmonary vein origins in this study. The authors wrote that in the Discussion section, but you should strongly state it in the Limitations section. In addition, the number of PV origin cases should also be clearly written in the manuscript.

Reviewer #3: Authors presented an interesting paper focused on the research of non PV trigger areas in the setting of atrial fibrillation.

PVI was firstly performed and induction of AF was subsequently attempted to observe non PVI foci.

Some considerations:

- retrospective analysis is not the best way to investigate the issue (this concept has been underlined in limitations)

- The possibility to discover extra PV trigger for AF as been previously well described (How many atrial fibrillation ablation candidates have an underlying supraventricular tachycardia previously unknown? Efficacy of isolated triggering arrhythmia ablation. Sciarra L, Rebecchi M, De Ruvo E, De Luca L, Zuccaro LM, Fagagnini A, Corò L, Allocca G, Lioy E, Delise P, Calò L.

Europace. 2010 Dec;12(12):1707-12. doi: 10.1093/europace/euq327. Epub 2010 Sep 10). Why has been not a first research of triggers attempted? Please add a comment.

6. PLOS authors have the option to publish the peer review history of their article (what does this mean?). If published, this will include your full peer review and any attached files.

Reviewer #1: No

Reviewer #2: No

Reviewer #3: **Yes: **Antonio Scarà

---

## [Author Response · Author response to Decision Letter 0]

19 Jan 2022

Reviewer #1: The work is very interesting and well written.

Thank you for your general understanding to our work.

I would discuss two aspects in depth:

1. Before the analysis of non-PV triggers, the importance of the electrophysiological study, especially in paroxysms without heart disease, should be highlighted in order to find any organized arrhythmias. Please read Sciarra L, Rebecchi M, De Ruvo E, De Luca L, Zuccaro LM, Fagagnini A, Corò L, Allocca G, Lioy E, Delise P, Calò L. How many atrial fibrillation ablation candidates have an underlying supraventricular tachycardia previously unknown? Efficacy of isolated triggering arrhythmia ablation. Europace. 2010 Dec;12(12):1707-12. doi: 10.1093/europace/euq327. Epub 2010 Sep 10. PMID: 20833693.

We completely agree with the reviewer. We well consider that SVT can be a trigger of AF in some cases. In the enrolled patients in the present study, no SVTs were induced except 3 cavotricuspid isthmus (CTI)-dependent atrial flutters in EPS. Now this is mentioned in the Result section. We have also added the mentioned study as a reference. The analyzed non-PV triggers in this study were ectopic foci that initiate fibrillatory activities from the beginning. 

(Results)

We considered supraventricular tachycardias (SVT) as possible triggers of AF as well [19]. CTI-dependent atrial flutters were induced during EPS in 3 cases, but no other SVTs were induced in the studied patients.

(References)

19. Sciarra L, Rebecchi M, De Ruvo E, De Luca L, Zuccaro LM, Fagagnini A, et al. How many atrial fibrillation ablation candidates have an underlying supraventricular tachycardia previously unknown? Efficacy of isolated triggering arrhythmia ablation. Europace. 2010;12:1707-12. 

2. I would also briefly mention the role of the substrate as the islets of fibrosis in the left atrium (Rillo M, Palamà Z, Punzi R, Vitanza S, Aloisio A, Polini S, Tucci A, Pollastrelli A, Zonno F, Anastasia A, Giannattasio CF, My L. A new interpretation of nonpulmonary vein substrates of the left atrium in patients with atrial fibrillation. J Arrhythm. 2021 Feb 22;37(2):338-347. doi: 10.1002/joa3.12521. PMID: 33850575; PMCID: PMC8021999. )

We completely agree with the reviewer and we also strongly recognize the relationship between left atrial degeneration and arrhythmogenic substrate. It is a　great theme whether to go with trigger mapping/ablation or substrate modification. In terms of AF trigger, we previously reported that non-PV AF trigger is likely to arise from degenerated atrial tissue where maintenance substrate of AF may co-exist. (reference #15)

Reviewer #2: Comments to the Author

This paper focused on the method to distinguish the areas of non-pulmonary vein triggers of AF with an electrode catheter.

I clinically agree with author’s ideas, but there are some major problems.

In particular, parts of the Methods section are considerably lacking.

We appreciate the reviewer’s general comment to our work. We would like to further explain our research design and what we would like to demonstrate. We considered the above-mentioned issues and revised the manuscript as below.

Major comments

1. This CS catheter you recommend is generally difficult to use. Aren’t the electrode catheters over-used, such as the PentaRay or BeeAT catheter? Therefore, I think it is difficult to use this method as a standardized algorithm.

As reviewer pointed out, the CS catheter (BeeAT) used in this study may not be available worldwide and we know that usable number of electrode is limited in some countries. What we intended to demonstrate in this study was that the basically-positioned catheters in the HRA and CS provide useful information to localize a non-PV trigger of AF and help further detailed mapping. Since we understand that our strategy cannot be universally recommended, we have revised and added sentences in Limitations section as below.

(Limitations)

From

Finally, the assessment of the intra-cardiac electrograms might be difficult soon after a defibrillation because it takes a few seconds for the baseline of electrograms to return to where it was.

To

Fourth, the assessment of the intra-cardiac electrograms might be difficult soon after a defibrillation because it takes a few seconds for the baseline of electrograms to return to where it was. Finally, the electrode catheter used in this study, which can simultaneously record CS and RA potentials (BeeAT), is not available worldwide and can only be used in limited regions. Therefore, it is necessary to verify whether the same results can be obtained using widely available separate CS and RA catheters.

2. Although there is a description of 751 sessions, the total number of cases and paroxysmal and non-paroxysmal cases are unknown. This study is a retrospective study and more patient information should be provided.

As the reviewer mentioned, the type of AF (paroxysmal and persistent) is very important in AF ablation studies. Therefore, the number of patients with paroxysmal and persistent AF and the number of non-PV triggers in all 751 sessions were added to the Methods as follows.

(Methods)

from

We retrospectively analyzed 37 non-PV triggers in 751 ablation sessions for symptomatic AF from January 2017 to December 2020.

To

We retrospectively analyzed 37 non-PV triggers in 751 ablation sessions for symptomatic AF (including 310 paroxysmal AF) from January 2017 to December 2020.

3. Were the 35 cases in this study completely cured for the entire follow-up period by the focal ablation of the non-pulmonary vein origins? Did you try to re-induce AF after the ablation of the non-pulmonary vein origins? This study found that paroxysmal and non-paroxysmal non-pulmonary vein origins were approximately equal. Moreover, in the intracardiac electrocardiogram shown in Fig. 1B, the onset of the AF is far from the previous QRS. From the above, it is hard to believe that it was the correct site to prove the origin of AF. Finally, I consider it to have an overall low credibility.

We appreciate the reviewer’s insightful comments. As for prognosis, we have previously reported that non-recurrence rate of successfully ablated non-PV trigger cases was comparable to those with no detectable non-PV triggers (reference #15). We would like the reviewer to understand that the present study is not a long-term outcome study but deals with a mapping methodology.

Yes. We re-induced AF after ablation to confirm the elimination of targeted non-PV triggers as stated in Methods. In order to certify the methodology, we added a sentence in Methods as follows.

(Methods)

AF induction test was repeated to see if there were any other non-PV triggers remaining.

In appreciation with the reviewer’s suggestion, we also have added sentences regarding the prevalence of non-PV triggers in patients with paroxysmal and non-paroxysmal AF respectively in the Discussion section as follows.

(Discussion)

The occurrence of non-PV triggers was comparable in paroxysmal (17 in 310, 5.5%) and in non-paroxysmal cases (18 in 441, 4.1%). The Prevalence of non-PV triggers was similar regardless of arrhythmia type, which is consistent to previous reports including ours [15, 23].

Finally, we would like to discuss the credibility of the non-PV mapping in Figure 1B. We would like to the reviewer to understand that Figure 1B is never showing an earliest activation electrogram but the measurement strategy in this study. As a note, coupling interval (CI) between the previous atrial potential and the ectopic beat potential recorded at the RA was 303 msec, which was not that long and the atrial activation showed so-called P on T. More than anything, the ectopic beat actually had initiated AF. We thus would like to ensure that atrial premature beat in Figure 1B is of an exact non-PV trigger.

4. There were too few cases of non-pulmonary vein origins in this study. The authors wrote that in the Discussion section, but you should strongly state it in the Limitations section. In addition, the number of PV origin cases should also be clearly written in the manuscript.

Thank you for this insightful comment. The incidence of non-PV triggers may be lower in this study than in some previous studies. However, please ensure that we just focused on atrial AF triggers and thus excluded SVC triggers in this study. Considering this, prevalence of non=PV triggers in this study is not that small. We think that SVC can be treatable in a preventive manner (as the 5th thoracic vein) and a SVC trigger is not difficult to map. What we intended to clarify was how to map an atrial trigger of AF. Prevalence of PV trigger in our EPS strategy is published in-detail in our previous study (refenrence #15). In this study, non-PV triggers were induced after completion of PVI as mentioned in Method. Therefore, non-PV triggers in the near PV antrum might have been eliminated by PVI. We now mentioned this point in Limitations.

(Limitations)

The low incidence in this study of non-PV triggers may be partly because we excluded SVC triggers in this study. Another reason may be because non-PV triggers were induced after completion of PVI in this study. Thus, non-PV triggers originating from near PV antrum might have been eliminated by PVI.

Reviewer #3: Authors presented an interesting paper focused on the research of non PV trigger areas in the setting of atrial fibrillation.

PVI was firstly performed and induction of AF was subsequently attempted to observe non PVI foci.

Thank you for your general understanding to our work.

Some considerations:

- retrospective analysis is not the best way to investigate the issue (this concept has been underlined in limitations)

As the reviewer mentioned, it is difficult to conclude only with a retrospective study. Therefore, we emphasized this point further in Limitations as below.

(Limitations)

From

First, this study was a retrospective study analyzing a limited sample size, therefore, a prospective study with more data is needed.

To

First, this study was a retrospective study analyzing a limited sample size, therefore, a prospective study with more data is needed to standardize this analysis method for clinical practice.

- The possibility to discover extra PV trigger for AF has been previously well described (How many atrial fibrillation ablation candidates have an underlying supraventricular tachycardia previously unknown? Efficacy of isolated triggering arrhythmia ablation. Sciarra L, Rebecchi M, De Ruvo E, De Luca L, Zuccaro LM, Fagagnini A, Corò L, Allocca G, Lioy E, Delise P, Calò L.

Europace. 2010 Dec;12(12):1707-12. doi: 10.1093/europace/euq327. Epub 2010 Sep 10). Why has been not a first research of triggers attempted? Please add a comment.

Thank you for this valuable comment. As the reviewer stated, SVT can be a cause of AF.　If a SVT could be induced during AF ablation procedure and EPS, we would definitely map and ablate it, whereas there were no cases in which SVT (AVNRT, AVRT, FAT) was induced after PVI in the present study. As in previous studies, there is a strategy to perform EPS eliciting the cause of AF before PVI, however, in order to reduce the recurrence of AF as much as possible, EPS was performed after PVI, which is a standard treatment for AF in practice. We now added the following sentences to the Result section. We have also added the mentioned study as a reference.

(Results)

We considered supraventricular tachycardias (SVT) as possible triggers of AF as well [19]. Cavotricuspid isthmus-dependent atrial flutters were induced during EPS in 3 cases, but no other SVTs were induced in the studied patients.

(References)

19. Sciarra L, Rebecchi M, De Ruvo E, De Luca L, Zuccaro LM, Fagagnini A, et al. How many atrial fibrillation ablation candidates have an underlying supraventricular tachycardia previously unknown? Efficacy of isolated triggering arrhythmia ablation. Europace. 2010;12:1707-12.

---

## [Decision Letter · Decision Letter 1]

9 Feb 2022

PONE-D-21-34225R1Intra-atrial activation pattern is useful to localize the areas of non-pulmonary vein triggers of atrial fibrillationPLOS ONE

Dear Dr. Mukai,

Thank you for submitting your manuscript to PLOS ONE. After careful consideration, we feel that it has merit but does not fully meet PLOS ONE’s publication criteria as it currently stands. Therefore, we invite you to submit a revised version of the manuscript that addresses the points raised during the review process.

ACADEMIC EDITOR: Dear Authors, after a careful evaluation of your paper, our opinion is that the quality of your manuscript significantly improved. However one of our reviewers is still suggesting some revisions. Best regardsLuigi Sciarra==============================

We look forward to receiving your revised manuscript.

Kind regards,

Luigi Sciarra

Academic Editor

PLOS ONE

Journal Requirements:

Additional Editor Comments (if provided):

Dear Authors, after a careful evaluation of your paper, our opinion is that the quality of your manuscript significantly improved. However one of our reviewers is still suggesting some revisions. Best regards

Luigi Sciarra

Reviewers' comments:

Reviewer's Responses to Questions

**Comments to the Author**

1. If the authors have adequately addressed your comments raised in a previous round of review and you feel that this manuscript is now acceptable for publication, you may indicate that here to bypass the “Comments to the Author” section, enter your conflict of interest statement in the “Confidential to Editor” section, and submit your "Accept" recommendation.

Reviewer #1: (No Response)

Reviewer #3: All comments have been addressed

2. Is the manuscript technically sound, and do the data support the conclusions?

Reviewer #1: Partly

Reviewer #3: Yes

3. Has the statistical analysis been performed appropriately and rigorously? 

Reviewer #1: Yes

Reviewer #3: Yes

4. Have the authors made all data underlying the findings in their manuscript fully available?

Reviewer #1: Yes

Reviewer #3: Yes

5. Is the manuscript presented in an intelligible fashion and written in standard English?

Reviewer #1: Yes

Reviewer #3: Yes

6. Review Comments to the Author

Reviewer #1: Point two on the role of extrapulmonary substrates has not been adequately investigated. I believe it is not possible to treat intra-atrial delays without talking about fibrosis!

Reviewer #3: (No Response)

7. PLOS authors have the option to publish the peer review history of their article (what does this mean?). If published, this will include your full peer review and any attached files.

Reviewer #1: **Yes: **Zefferino Palamà

Reviewer #3: **Yes: **Antonio Scarà

---

## [Author Response · Author response to Decision Letter 1]

17 Feb 2022

Reviewer #1: Point two on the role of extrapulmonary substrates has not been adequately investigated. I believe it is not possible to treat intra-atrial delays without talking about fibrosis!

Thank you for your excellent comment again about the atrial fibrosis.　To address the importance of the relationship between left atrial degeneration and arrhythmogenic substrate in AF ablation, the following sentence and the reference below were added in Discussion and Reference.

(Discussion)

For detailed mapping, it is also important to evaluate the substrate such as fibrosis by using multipolar electrodes [22].

(Reference)

22. Rillo M, Palamà Z, Punzi R, Vitanza S, Aloisio A, Polini S, et al. A new interpretation of nonpulmonary vein substrates of the left atrium in patients with atrial fibrillation. J Arrhythm. 2021;37:338-47.

---

## [Decision Letter · Decision Letter 2]

21 Feb 2022

Intra-atrial activation pattern is useful to localize the areas of non-pulmonary vein triggers of atrial fibrillation

PONE-D-21-34225R2

Dear Dr. Mukai,

We’re pleased to inform you that your manuscript has been judged scientifically suitable for publication and will be formally accepted for publication once it meets all outstanding technical requirements.

Kind regards,

Luigi Sciarra

Academic Editor

PLOS ONE

Additional Editor Comments (optional):

Reviewers' comments:

Reviewer's Responses to Questions

**Comments to the Author**

1. If the authors have adequately addressed your comments raised in a previous round of review and you feel that this manuscript is now acceptable for publication, you may indicate that here to bypass the “Comments to the Author” section, enter your conflict of interest statement in the “Confidential to Editor” section, and submit your "Accept" recommendation.

Reviewer #1: All comments have been addressed

2. Is the manuscript technically sound, and do the data support the conclusions?

Reviewer #1: Yes

3. Has the statistical analysis been performed appropriately and rigorously? 

Reviewer #1: Yes

4. Have the authors made all data underlying the findings in their manuscript fully available?

Reviewer #1: (No Response)

5. Is the manuscript presented in an intelligible fashion and written in standard English?

Reviewer #1: Yes

6. Review Comments to the Author

Reviewer #1: All queries have been properly addressed. The role of fibrosis in the genesis and maintenance of atrial fibrillation cannot be ignored

7. PLOS authors have the option to publish the peer review history of their article (what does this mean?). If published, this will include your full peer review and any attached files.

Reviewer #1: **Yes: **Zefferino Palamà

---

## [Editor Report · Acceptance letter]

13 Apr 2022

PONE-D-21-34225R2 

Intra-atrial activation pattern is useful to localize the areas of non-pulmonary vein triggers of atrial fibrillation 

Dear Dr. Mukai:

I'm pleased to inform you that your manuscript has been deemed suitable for publication in PLOS ONE. Congratulations! Your manuscript is now with our production department. 

Kind regards, 

on behalf of

Dr. Luigi Sciarra 

Academic Editor

PLOS ONE